# Differential impacts of freshwater and marine covariates on wild and hatchery Chinook salmon marine survival

**Brandon Chasco**[1]*, **Brian Burke**[2], **Lisa Crozier**[2], **Rich Zabel**[2]

**1** Fish Ecology Division, National Marine Fisheries Service, National Oceanic and Atmospheric Association, Newport, Oregon, United States of America, **2** Fish Ecology Division, National Marine Fisheries Service, National Oceanic and Atmospheric Association, Seattle, Washington, United States of America

☯ These authors contributed equally to this work.
* Brandon.Chasco@noaa.gov

**Data Availability Statement:** All relevant data are within the manuscript and the Supporting Information files.

## Abstract

Large-scale atmospheric conditions in the Northeast Pacific Ocean affect both the freshwater environment in the Columbia River Basin and marine conditions along the coasts of Oregon, Washington, and British Columbia, resulting in correlated conditions in the two environments. For migrating species, such as salmonids that move through multiple habitats, these correlations can amplify the impact of good or poor physical conditions on growth and survival, as movements among habitats may not alleviate effects of anomalous conditions. Unfortunately, identifying the mechanistic drivers of salmon survival in space and time is hindered by these cross-habitat correlations. To address this issue, we modeled the marine survival of Snake River spring/summer Chinook salmon with multiple indices of the marine environment and an explicit treatment of the effect of arrival timing from freshwater to the ocean, and found that both habitats contribute to marine survival rates. We show how this particular carryover effect of freshwater conditions on marine survival varies by year and rearing type (hatchery or wild), with a larger effect for wild fish. As environmental conditions change, incorporating effects from both freshwater and marine habitats into salmon survival models will become more important, and has the additional benefit of highlighting how management actions that affect arrival timing may improve marine survival.

## Introduction

Snake River spring/summer Chinook salmon are an iconic species of the Pacific Northwest. Populations once supported large commercial and recreational fisheries, as well as subsistence for indigenous communities. However, their complex life cycle leaves them vulnerable to the influences of climate and climate change at several life stages [1]. Further, the correlation of environmental conditions across space and time can exacerbate this vulnerability. Indeed, recent research suggests that freshwater effects carry over into the marine realm and may hinder recovery [2–5], but recent applications of generalized linear models to answer this question do not account for the random variability in the carry-over process [6,7]. These simplifications

**Funding:** There were no grants or funders associated with this work. All authors were paid from their NOAA government salaries. The funders had no role in study design, data collection and analysis, decision to publish, or preparation of the manuscript. We did not receive any funding or salary from an outside agency or institution.

**Competing interests:** The authors have declared that no competing interests exist.

of complex processes in salmon survival models may lead to biases in the parameter estimates and narrow estimates of the standard errors for parameters and derived variables [8], which may compromise our ability to make robust forecasts under future environmental conditions and management scenarios.

Chinook salmon are a semelparous fish with a complex life history, and their survival depends on processes in both freshwater and marine environments over thousands of kilometers [9]. The majority of spring/summer Chinook salmon in the Snake River ESU spend two years in freshwater from the adult spawning migration to the juvenile outmigration and two years in the ocean, with ocean survival showing strong dependence on climatic conditions [10,11]. Data from juvenile Chinook salmon uniquely tagged in the freshwater environment and detected as returning adults suggest that the period when salmon first enter the marine environment is critical to overall marine survival [12–14]. Unfortunately, many of the specific mechanisms of mortality during this period are not well known.

Evaluating drivers of survival for migrating animals is difficult because the interaction between physical processes at local, regional, and basin scales commonly results in correlated conditions across nearby habitat types. In the Columbia River Basin, inter-annual variability in freshwater conditions tends to be correlated with variability in regional marine conditions [7,15], as both habitats are driven by large-scale atmospheric and oceanographic forces. This correlation has the potential to amplify (or dampen) anomalous conditions in multiple habitats simultaneously, thus complicating our ability to identify causative mechanisms of variability in salmon survival [16].

Carryover effects are effects that "carry over" from one life stage to another [17]. We propose the working definition: in an ecological context, carryover effects occur in any situation in which an individual's previous history and experience influences their current performance. While effects such as length, weight, and freshwater environmental covariates may be explored in future analyses [7], we focused our attention on how phenology (i.e., migration timing) in the freshwater environment (the previous history) carries over to the survival in the marine environment (the current life stage). In part, we focused our attention on the phenology because of the emphasis on climate change affecting freshwater conditions in natural systems (e.g., reduced stream flows, warmer water temperatures [18]) and the practical application for co-managers in highly regulated systems (e.g., the upper Columbia River and Snake River basins) where restoration of the natural migration conditions for juvenile salmon has been a priority. Given the controversy over juvenile transportation and the economic cost of spilling water over hydroelectric dams [18], avoiding the imposition of a fixed form of this relationship is especially important.

Previous estimates of smolt-to-adult returns (SAR) used either generalized linear models (glm) and treated the temporal variability in survival with fixed effects for day, $day^2$, and the day/year interaction [6], or generalized mixed effects models (glmm) with day and or year as independent uncorrelated random effects [7]. Numerous research efforts have shown that not accounting for autocorrelation in fisheries data can lead to biases in parameters estimates and derived variables of the models [19], and it is unknown whether the fixed effects for day and day/year interactions can lead to biases in the expectation and uncertainty in salmon survival models.

In this effort, we provide a generalized statistical model that scientists and managers can use to integrate the complex interacting effects of environmental conditions across multiple habitats with estimates of salmon survival. From a modeling perspective, our justification for including autocorrelated random effects for the day, year, and day/year interactions recognizes that natural systems have inherent structure in the variability that a fixed effect model is unlikely to capture, and a random effects model allow us to decouple the uncertainty in the

day effect from the uncertainty in the observations. From a biological perspective, other factors not measured during the migration period are also likely to affect salmon survival (e.g., predation) and therefore the effect of migration day may not be as smooth as a quadratic curve that is constant among years. Our model is meant to provide a parametric estimate of the autocorrelation at the daily and annual time scales, recognizing that these data were collected over time and the independence between observations is likely to be affected by the duration between observations. To evaluate our approach relative to previous models of salmon survival, we applied a random effects model to a rich dataset of over 285,000 individually-tagged Snake River spring/summer Chinook salmon between 2000 and 2015.

## Methods

### Fish data

We used Passive Integrated Transponder (PIT) data provided by Columbia Basin Research (CBR, cbr.washington.edu) via PIT Tag Information Systems (PTAGIS, www.ptagis.org) to estimate the survival of juvenile salmon. We considered data for all out-migrating yearling spring/summer Chinook salmon tagged in the Snake River Basin detected from 2000 to 2015 at Bonneville Dam—the furthest downstream dam on the Columbia River. We marked a fish as having survived the marine stage if it was detected at Bonneville Dam as an adult. We also included detections farther upstream for the less than 2% (5,712 out of 285,600) of fish that were missed at Bonneville Dam. The data included i) last detection date at Bonneville Dam as juveniles, ii) rear type (hatchery or wild), and iii) whether the fish was detected in the Columbia River as an adult. We excluded all fish with an unknown rearing type (i.e., hatchery versus wild), geographic regions with fewer than 200 individuals (over the 16 years), those fish released or tagged below the confluence of the Snake and Columbia Rivers, fish that returned to spawn in less than one year, and fish that did not volitionally migrate (i.e., placed into barges to avoid passage through the hydrosystem) downstream as juveniles. Additionally, we excluded fish that passed Bonneville Dam prior to April 9th (day 100) or later than July 8th (day 190); these fish account for <0.14% of the total observations. There is very little data to inform the temporal autocorrelation at the margins of the migration period, and an initial analysis demonstrated that removing these observations greatly improved the speed and convergence of the model fit with little change in the estimates of model parameters. In total, there were 285, 244 individuals for this analysis (Table 1). All PIT-tag files are available on the CBR website (http://www.cbr.washington.edu/dart/cs/data/nmfs_sar/).

**Table 1. Sample sizes for spring/summer Chinook salmon.**

| Watershed | Hatchery | Wild |
|---|---|---|
| Clearwater | 88,895 | 4,512 |
| Grande Rhonde and Wallowa | 16,956 | 5,636 |
| Imnaha, Pahsimeroi, South Fork Salmon | 60,215 | 14,622 |
| Little Salmon | 69,016 | 8 |
| Lower Snake and Tucannon | 11,125 | 988 |
| Middle Fork Salmon | 0 | 3,714 |
| Upper Salmon River (above Yankee Fork) and Lemhi | 5,589 | 4,384 |
| Total | 251,796 | 33,864 |

Sample size of hatchery and wild juvenile Chinook salmon from different watersheds within the Snake River Basin.

## Environmental data

Because early ocean experiences are thought to have a large influence on salmon ocean survival [3–5], we focused environmental correlates on marine conditions spanning the winter prior to when fish out-migrated to the fall after outmigration. We obtained these environmental covariate data from a variety of sources (Table 2). Variables represent large-scale oceanographic

**Table 2. Description of model covariates.**

| Variable | Description | Years Available | URL / Source |
|---|---|---|---|
| CRflow.spr<br>CRflow.sum | Seasonal Columbia River flow as measured at The Dalles Dam (USGS site 14105700) | 1978-present | http://waterservices.usgs.gov/rest/DV-Service.html |
| CRtemp.spr<br>CRtemp.sum | Seasonal Columbia River temperatures at The Dalles Dam (USGS site 14105700) | 1997-present | http://waterservices.usgs.gov/rest/DV-Service.html |
| cui.win<br>cui.spr<br>cui.sum<br>cui.aut | Seasonal coastal upwelling index | 1946-present | http://www.pfeg.noaa.gov/products/PFELData/upwell/monthly/upanoms.mon |
| mei.win<br>mei.spr<br>mei.sum<br>mei.aut | Seasonal Multivariate ENSO Index | 1950-present | https://www.esrl.noaa.gov/psd/enso/mei/ |
| npgo.win<br>npgo.spr<br>npgo.sum<br>npgo.aut | Seasonal North Pacific Gyre Oscillation | 1950-present | http://www.o3d.org/npgo/npgo.php |
| npi.win<br>npi.spr<br>npi.sum<br>npi.aut | Seasonal North Pacific Index (index of Aleutian Low Pressure) | 1899-present | https://climatedataguide.ucar.edu/sites/default/files/npindex_monthly.txt |
| oni.win<br>oni.spr<br>oni.sum<br>oni.aut | Seasonal Oceanic Niño Index | 1950-present | http://www.cpc.ncep.noaa.gov/products/analysis_monitoring/ensostuff/ensoyears.shtml |
| ersstWACoast.win<br>ersstWACoast.spr<br>ersstWACoast.sum<br>ersstWACoast.aut | Seasonal sea surface temperature for coastal Washington | 1900-present | https://www1.ncdc.noaa.gov/pub/data/cmb/ersst/v5/netcdf/ |
| ersstArc.win<br>ersstArc.spr<br>ersstArc.sum<br>ersstArc.aut | Seasonal sea surface temperature from Johnstone and Mantua (2014) | 1900-present | https://www1.ncdc.noaa.gov/pub/data/cmb/ersst/v5/netcdf/ |
| transport.win<br>transport.spr<br>transport.sum<br>transport.aut | Seasonal measure of Sverdrup transport along the Washington coast, most correlated with the temperatures in the upper 20 meters. | 1967-present | |

Description of the environmental variable names, the years of available data, and the website location of the data.

patterns as well as regional physical metrics. While not all variables have a proposed direct mechanistic relationship with salmon survival, they have been shown to correlate with Chinook salmon returning to specific ESUs [20,21]. The environmental variables in our models include basin-scale oceanic and atmospheric variables (i.e., North Pacific Gyre Oscillation (PGO), Oceanic Niño Index (ONI), Multivariate ENSO Index (MEI), and North Pacific Index (NPI)), local or regional variables (i.e., sea surface temperature for coastal Washington (ersstWACoast), sea surface temperature from Johnstone and Mantua (2014) (ersstARC), coastal upwelling index (CUI), a measure of the Sverdrup index that is correlated with the temperatures in the upper 20 meters (transport)), and indicators from the Columbia River representing the environment that salmon inhabited just prior to migrating into the ocean (i.e., Columbia River flow (CRflow) and Columbia River temperature (CRtemp)). Furthermore, we binned all environmental data into three-month averages: these seasonal metrics include Dec-Feb ('win'), Mar-May ('spr'), Jun-Aug ('sum'), and Sep-Nov ('aut'). These seasonal bins are identified as suffixes on the environmental data names. For all of the marine variables included in our analyses, we tested each of the four seasons, starting with the winter prior to when salmon enter the ocean.

### Estimation and data processing

All of the data we used for this analysis are publicly available. We provide a description of the R scripts used to create these environmental data objects from the raw data inputs in S1 Table. The estimation of the model parameters was done with Template Model Builder (TMB)–a package of C++ libraries that efficiently estimates fixed effects of the model using the AutoDiff libraries and a Laplace approximation to integrate over the random effects.

### Models

We used a mixed-effect logistic regression model to predict the SAR for fish of each rear type (i.e., hatchery versus wild) migrating past Bonneville Dam on calendar day $j$ during year $t$. The hatchery and wild fish data were modeled separately, as such our model does not include a subscript for rear type.

$$s_{jt} = \frac{e^{\eta_{jt}}}{1 + e^{\eta_{jt}}} \tag{1}$$

$$\eta_{jt} = \mu + \boldsymbol{\beta}\mathbf{x} + v_j + \omega_t + \xi_{jt} \tag{2}$$

Where the link function $\eta_{jt}$ is a linear combination of mean survival, $\mu$, and $\boldsymbol{\beta}$ for the vector of fixed effect coefficients corresponding to the matrix of marine variables $\mathbf{X}$, plus random effects of $v_j$ for the day effect, $\omega_t$ for the year effect, and $\xi_{jt}$ for the interaction between calendar day $j$ and year $t$. A complete list of the subscripts, parameters and data are available in Table 3.

Given the total number of juveniles that migrated downstream $n_{jt}$ and the predicted SAR $s_{jt}$, the number of juvenile fish $k_{jt}$ that survived to adulthood and were detected at one of the eight main-stem on the Columbia River and Snake River is binomially distributed

$$k_{jt} \sim \text{Binomial}(n_{jt}, s_{jt}) \tag{3}$$

The random effect for calendar day and year were treated as auto-regressive processes with lag 1 (i.e., AR(1)),

$$v_j \sim \text{Normal}\left(\tau v_{j-1}, \frac{\psi^2}{1 - \tau^2}\right) \tag{4}$$

**Table 3. Description of model symbols.**

| Type | Symbol | Description |
|---|---|---|
| Data | $\eta_{jt}$ | cohort of juvenile fish migrating past Bonneville Dam on day $j$ in year $t$ |
| | $k_{jt}$ | number of fish from the juvenile cohort migrating past Bonneville Dam on day $j$ in year $t$ that survived to adulthood |
| | $\mathbf{X}$ | matrix of environmental covariates for each year |
| Index | $j$ | calendar day |
| | $t$ | year |
| | $\delta$ | Number of days between $j$ and $j + \delta$ |
| Fixed-effects | $\mu$ | mean survival |
| | $\boldsymbol{\beta}$ | vector of marine covariate parameters |
| | $\tau$ | correlation coefficient for day effect |
| | $\pi$ | correlation coefficient for the year effect |
| | $\rho$ | correlation coefficient for the day effect in the day/year interaction |
| | $\gamma$ | correlation coefficient for the year effect in the day/year interaction |
| | $\psi^2$ | variance of the day effect |
| | $\sigma^2$ | variance of the day year interaction |
| Random effects | $v_j$ | day effect for day $j$ |
| | $\omega_t$ | year effect for year y |
| | $\xi_{jt}$ | day/year effect for day $j$ and year $t$ |

List of data types, subscripts, parameters (i.e., fixed effects), and random effects used to model the smolt to adult survival of spring/summer Chinook salmon originating from the Snake River Basin.

$$\omega_t \sim \text{Normal}\left(\pi\omega_{t-1}, \frac{\varphi^2}{1-\pi^2}\right) \tag{5}$$

where, $\tau$ and $\pi$ are the correlations and $\frac{\psi^2}{1-\tau^2}$ and $\frac{\varphi^2}{1-\pi^2}$ are the variances, respectively. The random effect for the interaction between day and year was treated as a two-dimensional auto-regressive process,

$$\xi_t \sim \text{MVN}(\gamma\xi_{t-1}, \boldsymbol{\Sigma}) \tag{6}$$

where, $\xi_t$ is a vector of random effects across calendar days in year $t$, $\gamma$ is the correlation of the vector of day effects between years $t$ and $t$-$1$, and $\Sigma$ is the covariance matrix between days within a year. The covariance matrix $\Sigma$ is a compact way of representing the covariance for the day effects in the day/year interaction.

$$\Sigma_{j,j+\delta} = \sigma^2 \frac{\rho^\delta}{1-\rho^2} \tag{7}$$

Where the elements of the covariance matrix ($\Sigma$) are a function of the variance parameter $\sigma^2$ which is rescaled based on the correlation $\rho$ between days and the $\delta$ number of days between observations.

To estimate the fixed and random effects of the model, we use the non-linear optimization libraries in Template Model Builder package [22] built for R [23]. The marginal likelihood of the vector of fixed effects ($\boldsymbol{\theta}$) and the variance parameters ($\boldsymbol{\kappa}$) for the random effects ($\epsilon$) given the data (L[Data]) is maximized by integrating across the product of the conditional

probability of the data given the fixed and random effects (Pr ($\boldsymbol{\theta},\epsilon$)), and the probability of the random effects and the estimated variances (Pr ($\boldsymbol{\kappa}$); Thorson & Minto 2014),

$$L[\text{Data}] = \int_\epsilon \Pr(\boldsymbol{\theta}, \epsilon)\Pr(\boldsymbol{\kappa})d\epsilon \qquad (8)$$

Not all model combinations may be estimable due to the confounding effects among model parameters; in some instances, more than one model parameterization may produce identical fits to the data. In these cases, the Hessian is non-positive definite, and the solution is not unique or estimable. We define a converged model as one with a positive definite Hessian and a maximum gradient of 0.001 for the fixed effects. To compare models and select the most parsimonious fit to the data, we used the marginal AIC for the fixed effects (Akaike's information criterion; [24]) using the TMBhelper package.

Testing all of the thousands of parameter combinations for the 31 marine variables, in addition to the different combinations of random effects, is not reasonable. We therefore restricted the potential models to only those with i) zero, one, or two marine covariates and ii) only two-covariate models where the correlation between covariates was less 0.7. Furthermore, initial analyses indicated that estimating random effects for day, year, and the day/year interaction in a single model produced an over-fit to the data. Models with all three random effects did converge in some instances, but the magnitude of random effects for either the day or year were so small (<1e-4 in most cases) as to be meaningless. Therefore, we restricted our analysis to no more than two random processes for day, year, and the day/year interaction. This resulted in six different random effect models. Finally, to allow for the most flexibility for a given group of fish, we did not combine the hatchery and wild datasets in a multivariate analysis, but rather ran models for each dataset separately. Further research examining the covariance of these two groups could be considered in future analyses.

## Model validation

To further insure that the parameters of the random effects model ($\boldsymbol{\theta}$ and $\boldsymbol{\kappa}$) are estimable and unbiased over a range of biological conditions, we conducted simulations in a three by three factorial design—three separate trials for three different simulation experiments comparing the effects of sample size and auto-correlation in the random processes (see S1 Table for a list of simulation R scripts). The first experiment set the simulated sample sizes equal to 50%, 100%, and 500% of the observed sample sizes. The second experiment examined the correlation for the day effect by fixing $\tau$ to 0.1, 0.5, or 0.9. While the third experiment examined the effects of the correlations for the day/year interaction by fixing both $\rho$ and $\gamma$ equal to 0.1, 0.5, or 0.9. For each trial we generated 500 random data sets of the number of wild smolts that survived to adulthood for each day and year in our study based on the unconditional likelihood (i.e., simulated observations were generated based on uncertainty in the observation and random processes). We then compared the true parameters ($\boldsymbol{\theta}$ and $\boldsymbol{\kappa}$) to estimated parameters for the $k^{\text{th}}$ simulated data from trial $i$ and experiment $h$ ($\hat{\boldsymbol{\theta}}_{kih}$ and $\hat{\boldsymbol{\kappa}}_{kih}$). Finally, we compared the bias and precision of the model parameters when a fixed effect model with a quadratic term for migration day was fit to simulated data from the unconditional likelihood for the model with the most parsimonious fit to the wild salmon data.

Additionally, we used the area under the curve (AUC) statistic based on the receiver operator characteristic (ROC) graphs in the R package pROC [25]. The AUC statistic summarizes the model's ability to discriminate between true positive and false positive rates for a range thresholds. For ecological models, AUC values below 0.7 suggest poor discrimination in the model, values between 0.7 and 0.8 suggest an acceptable level of discrimination, and values greater than 0.9 implies the models provide excellent discrimination [26].

## Results

### Model Fit

We found that for wild fish, the models with random effects for day and day/year interactions along with two marine covariates produced the most parsimonious model fit to the data, and for hatchery fish, models with only day/year interactions and two environmental covariates produced the most parsimonious fit (Table 4). The top models (ΔAIC≤4) for wild fish all assumed random effects for day and day/year interactions, with differences in model fit arising from the combinations of marine covariates (Table 4). A small set of covariates informed the top models for hatchery fish and there was little evidence for an underlying day effect: the only top model for hatchery fish with a day effect had a ΔAIC equal to 4.

Comparing the most parsimonious model fits for each rearing type, our results suggested that the expected survivals and 95% confidence intervals for wild and hatchery fish were 0.009 (0.002, 0.035) and 0.008 (0.006, 0.010), respectively (Table 5). The marine covariates that improved the fit of the survival model were different for wild and hatchery fish, but the magnitude of the environmental effects was similar for the two rearing types (Table 5, Fig 1). Spring coastal upwelling index (cui.spr) and summer Pacific decadal oscillation (pdo.sum) provided the most parsimonious fit to the wild fish data, while summer transport (transport.sum—a measure of the northward transport of water based on the Sverdrup index) and the summer north Pacific gyre oscillation index (npgo.spr) provided the most parsimonious fit to the hatchery fish data. The percent change in marine survival as a function of the marine covariates varied between -70% to 150% for wild fish, and -70% and 200% for hatchery fish (Fig 1).

Across all of the top models (ΔAIC<4), we found differences in the importance of the marine covariates that explained hatchery and wild survival based on AIC weights (see Fig 2 for calculation). For the top models listed in Table 4, the coastal upwelling index (CUI),

**Table 4. Model comparison for hatchery and wild spring/summer Chinook salmon.**

| Rear type | ΔAIC | Random effect | | | Covariate | | | |
| | | Day ($\nu_{rj}$) | Year ($\omega_{rt}$) | Day/Year ($\varepsilon_{rjt}$) | 1st | 2nd | Gradient | PD Hessian |
|---|---|---|---|---|---|---|---|---|
| Hatchery | 0.0000 | N.E. | N.E. | X | transport.sum | npgo.sum | 0.00035 | TRUE |
| | 1.0002 | N.E. | N.E. | X | transport.sum | npgo.aut | 0.00175 | TRUE |
| | 1.2726 | N.E. | N.E. | X | transport.sum | npgo.spr | 0.00116 | TRUE |
| | 2.9505 | N.E. | N.E. | X | ersstArc.win | transport.sum | 0.00000 | TRUE |
| | 3.0477 | N.E. | N.E. | X | ersstArc.spr | transport.sum | 0.00070 | TRUE |
| Wild | 0.0000 | X | N.E. | X | cui.spr | pdo.sum | 0.00019 | TRUE |
| | 1.7894 | X | N.E. | X | ersstArc.spr | cui.spr | 0.00000 | TRUE |
| | 1.8590 | X | N.E. | X | npi.sum | pdo.spr | 0.00029 | TRUE |
| | 2.0080 | X | N.E. | X | ersstArc.win | ersstWAcoast.sum | 0.00007 | TRUE |
| | 2.0683 | X | N.E. | X | ersstWAcoast.sum | pdo.win | 0.00007 | TRUE |
| | 2.1123 | X | N.E. | X | ersstArc.win | cui.spr | 0.00002 | TRUE |
| | 2.4955 | X | N.E. | X | cui.spr | pdo.spr | 0.00009 | TRUE |
| | 3.2846 | X | N.E. | X | ersstWAcoast.sum | pdo.spr | 0.00010 | TRUE |
| | 3.3064 | X | N.E. | X | ersstWAcoast.sum | oni.spr | 0.00028 | TRUE |
| | 3.5296 | X | N.E. | X | ersstArc.win | npi.spr | 0.00018 | TRUE |
| | 3.5882 | X | N.E. | X | ersstWAcoast.spr | npi.sum | 0.00030 | TRUE |
| | 3.9492 | X | N.E. | X | cui.spr | pdo.win | 0.00001 | TRUE |

Top models for each origin type based on the random effects and number of marine covariates. The gradient is a measure of the likelihood surface for the maximum likelihood estimate, PD Hessian stands for positive definite Hessian, and N.E. stands for not estimated.

Washington coastal and arc sea surface temperatures (ersstWAcoast and ersstArc, respectively), and Pacific decadal oscillation (PDO) were important for wild fish (Fig 2), while transport and North Pacific gyre oscillation (NPGO) were most important for hatchery fish. Aside from spring upwelling, covariate indices in summer had stronger weight than other seasons (Fig 2).

For wild fish there was consistently higher survival for the earlier arriving fish—hence, the model with the lowest AIC included the random effect for day (Table 4). The interaction between day and year was important in the most parsimonious model fits for both wild and hatchery rearing types (Tables 4 and 5). Differences in the estimated daily survival rates varied from 0.002 to 0.115 for wild fish, and from 0.003 to 0.06 for hatchery fish (Fig 3). For the day/year effect on the survival of wild fish, there was a strong positive correlation among days within a year ($\rho = 0.932$), and negative correlation among days across years ($\pi = -0.489$) (Table 5). The random deviation of the day/year interaction for hatchery fish showed a high degree of correlation among days within a year ($\rho = 0.955$) and a weak negative correlation among days across years ($\pi = -0.067$). The standard deviation of the day/year interactions was similar for hatchery fish ($\varphi = 0.611$) and wild fish ($\varphi = 0.58$).

To illustrate the effect of arrival timing for wild and hatchery fish, we compared the top model for each rearing type that included the random effects for both the day and the day/year interactions. For wild fish, this was the model with the lowest AIC, and for hatchery fish, this was a model with identical marine covariates to the most parsimonious model but with daily random effects ($\Delta AIC = 4$; Table 4). The day effect was highest for wild fish passing Bonneville Dam around May 3$^{rd}$, followed by decreasing survival throughout the remainder of the smolt migration (Fig 4A). By comparison, the model of hatchery fish that included both day and day/year interaction showed no real difference in smolt survival for the day effect (Fig 4A), despite relatively similar mean arrival timing past Bonneville Dam (Fig 4B). The lack of a day effect for hatchery fish is supported by the low estimates for the correlation coefficient and variability in their day effect ($\tau = 0.05$ and $\psi = 0.134$). Conversely, the wild fish had higher correlation and variability ($\tau = 0.986$ and $\psi = 0.793$, respectively) which suggests that the day effect "wanders" more for wild fish.

While none of the top models included a random deviate for year, we predicted the annual survival by aggregating the daily survival estimates weighted by the total number of hatchery

**Table 5. Estimated models parameters.**

| Parameter description | Symbol | Hatchery | | Wild |
| --- | --- | --- | --- | --- |
| | | without day effect | with day effect | |
| Mean annual survival | $\mu$ | 0.008 (0.006, 0.01) | 0.008 (0.006, 0.011) | 0.009 (0.002, 0.035) |
| Correlation of day effect | $\tau$ | N.E. | 0.05 (0.003, 0.517) | 0.986 (-0.992, 0.994) |
| Correlation of day in day/year effect | $\rho$ | 0.955 (0.853, 0.972) | 0.958 (0.835, 0.974) | 0.932 (0.241, 0.963) |
| Correlation of year in day/year effect | $\pi$ | -0.067 (-0.355, 0.248) | -0.109 (-0.397, 0.229) | -0.489 (-0.707, 0.058) |
| Process error for day effect | $\psi$ | N.E. | 0.134 (0.02, 0.905) | 0.793 (0.243, 2.588) |
| Process error for day/year effect | $\varphi$ | 0.58 (0.449, 0.749) | 0.576 (0.442, 0.75) | 0.611 (0.451, 0.826) |
| Effect of first marine covariate[1] | $\beta_1$ | 0.488 (0.228, 0.747) [1] | 0.478 (0.218, 0.737) [2] | 0.458 (0.22, 0.695) [3] |
| Effect of second marine covariate[1] | $\beta_2$ | 0.547 (0.283, 0.812) [1] | 0.56 (0.295, 0.825) [2] | -0.608 (-0.82, -0.396) [3] |

[1]The first and second marine covariates are elements of the covariate vector **β**.

[2]For hatchery spring/summer Chinook salmon, the first and second marine covariates in the model with the lowest AIC were transport.sum and npgo.sum, respectively.

[3]For wild spring/summer Chinook salmon, the first and second marine covariates in the model with the lowest AIC were cui.spr and pdo.sum, respectively.

Estimated fixed effects for the model with lowest marginal AIC values for wild spring/summer Chinook salmon, and the models with the lowest marginal AIC for hatchery fish with and without a random day effect included (N.E. means not estimated).

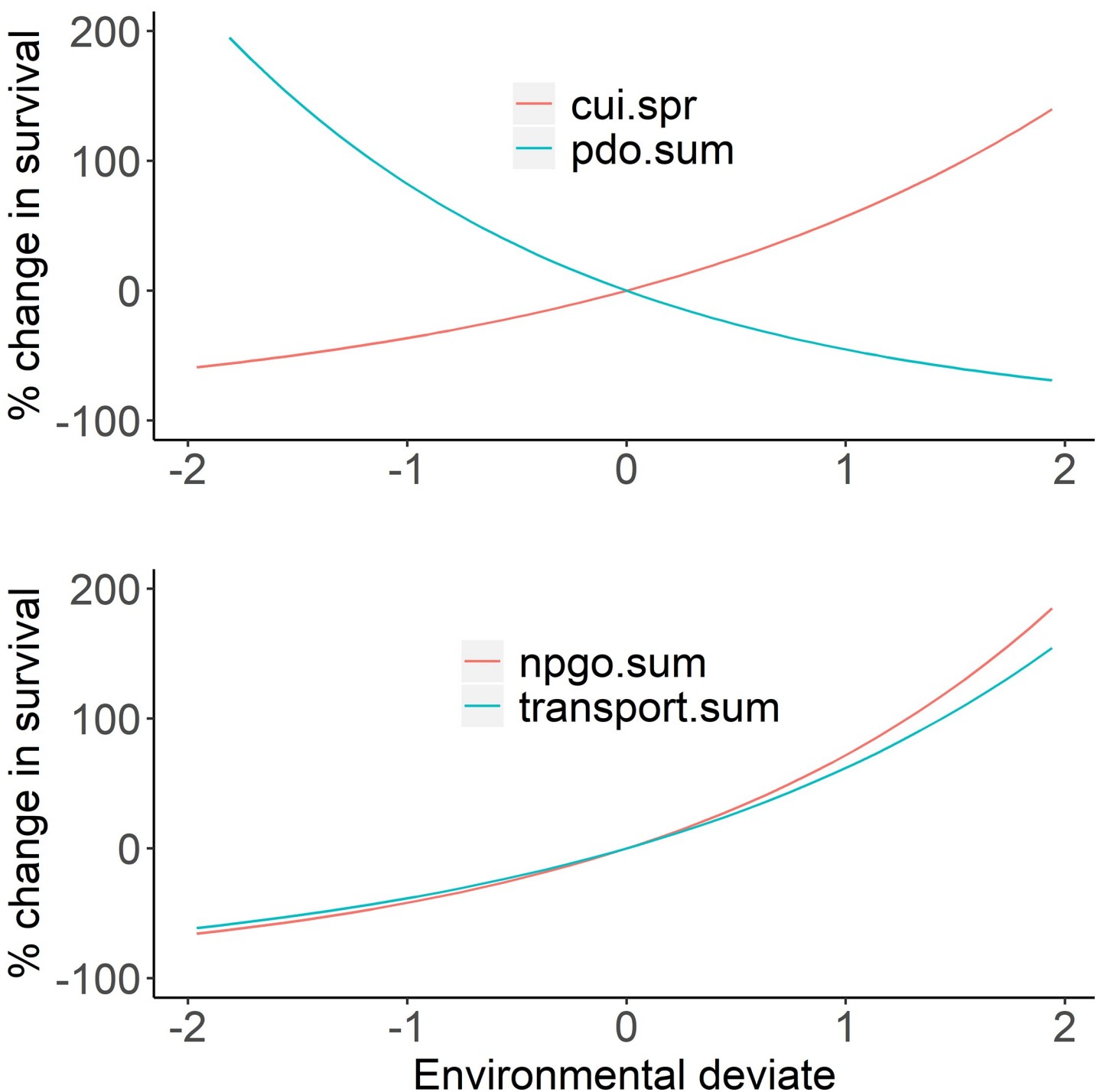

**Fig 1. Effects of environmental covariates on spring/summer Chinook salmon survival.** Environmental effects on survival of wild (upper panel) and hatchery (lower panel) spring/summer Chinook salmon based on the model fit to the observed data as selected by AIC (see Table 4 for summary of model fits).

and wild fish that arrived each day. The observed annual survival estimates were similar to the model predictions and, with the exception of wild fish in 2003, the observations fell within the 95% confidence interval (Fig 5). Both the predicted and observed annual survivals for hatchery

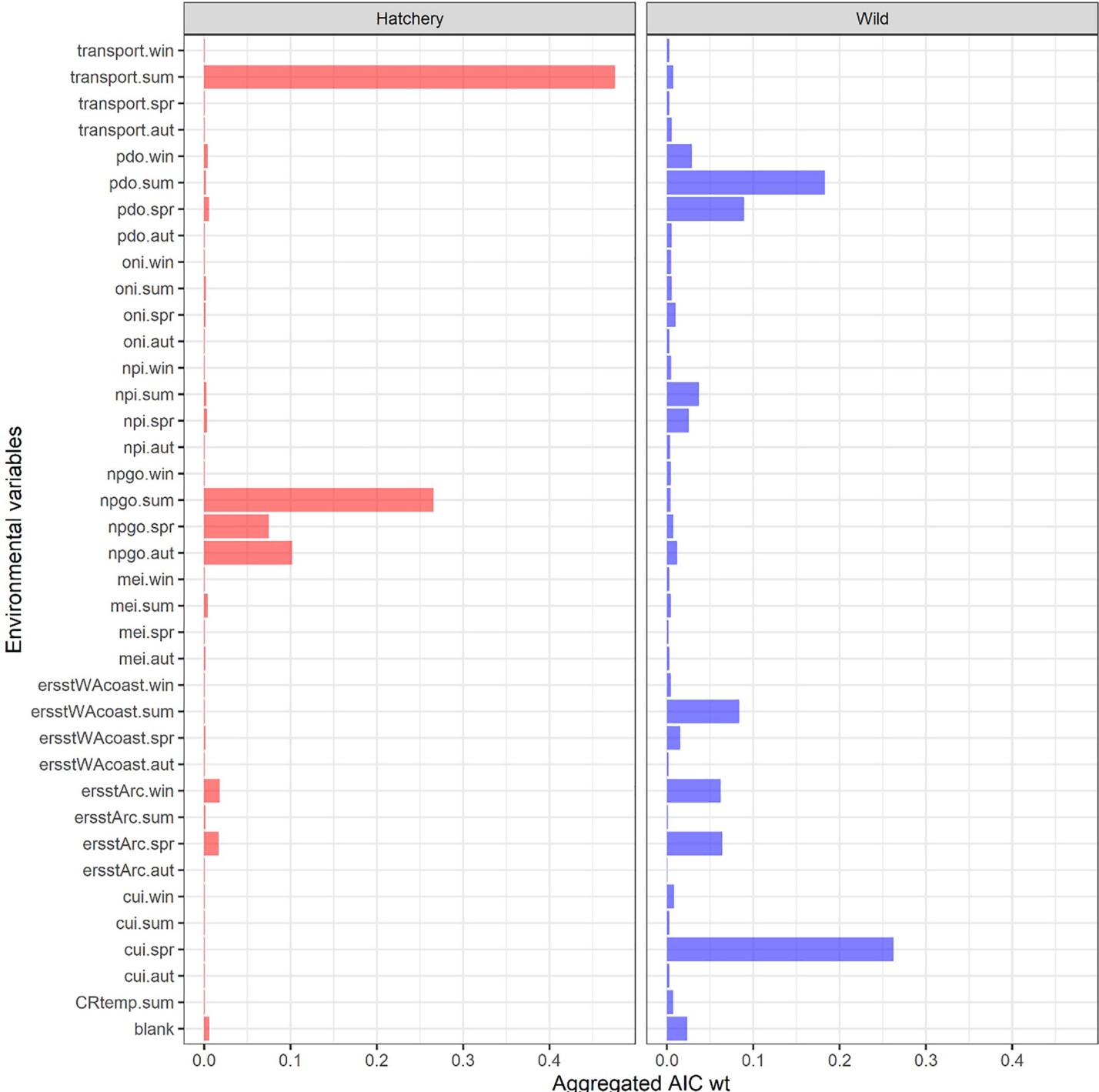

**Fig 2. Relative importance of environmental covariates in spring/summer Chinook salmon survival models.** Relative importance of the different marine covariates for predicting the marine survival of hatchery (left column) and wild (right column) spring/summer Chinook salmon, where the aggregated weight of a covariate c is equal to the sum of the AIC weights for all $m$ models containing covariate $c$, divided by the total weight across all $m$ models $\left( \frac{\sum_{c \in m} e^{\Delta AIC_m}}{\sum_m e^{\Delta AIC_m}} \right)$. The "blank" environmental variable is for models with no environmental predictors.

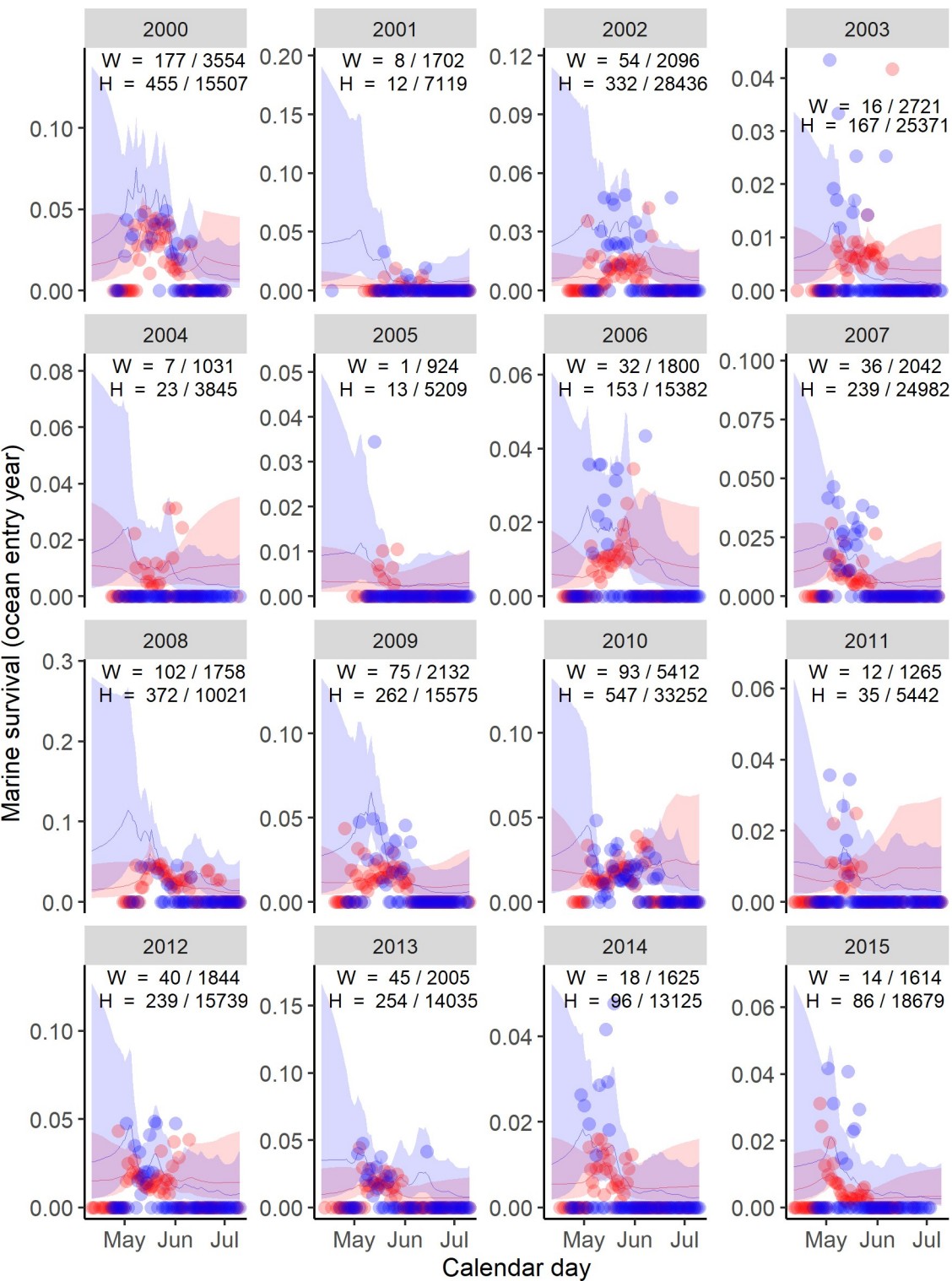

**Fig 3. Predicted survival of spring/summer Chinook salmon from the Columbia River.** The observed (dots), and maximum likelihood estimates (line) with 95% confidence intervals (ribbons) for the marine survival wild (blue) and hatchery (red) origin Spring/Summer Chinook salmon past Bonneville dam from 2000 to 2015. Each point represents the mean survival of all fish detected at Bonneville Dam on a particular day and year. Annual samples sizes of the survivors and total PIT tagged hatchery (H) and wild (W) for are shown in each panel. To maintain the readability of individual panels, mean observed survivals greater than 0.2 are not plotted.

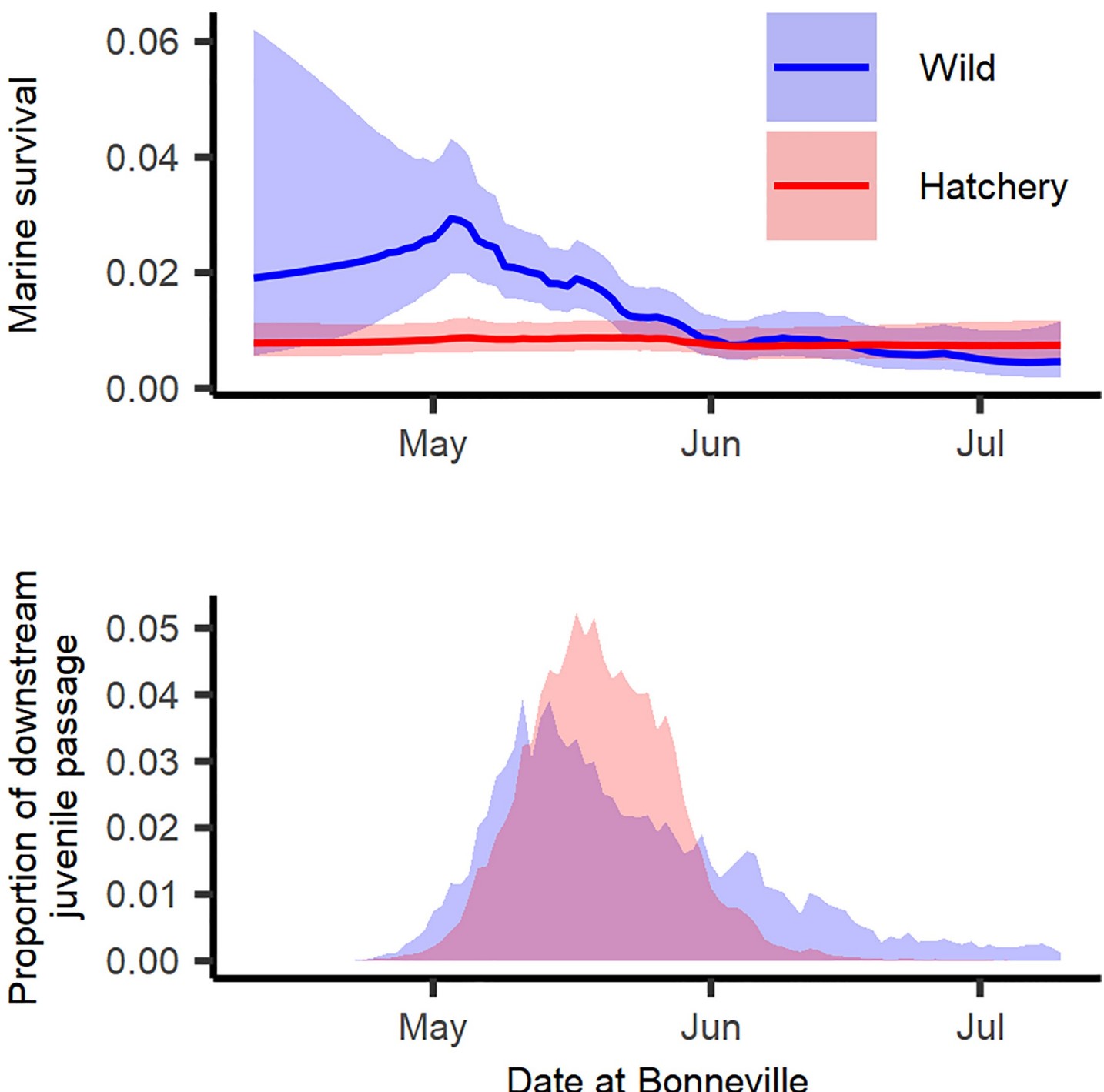

**Fig 4. The effect of migration day on the survival of spring/summer Chinook.** Predicted smolt-to-adult survival by day for hatchery (red) and wild (blue) spring/summer Chinook salmon (upper panel) for the most parsimonious model fits for each rear type that include both day and day/year interactions (see Table 4). Lines represent expected survivals and shaded regions represent 95% confidence intervals. Average daily proportion (across all years) of smolts arriving to and migrating past Bonneville Dam (2000 to 2015) (lower panel).

and wild fish showed an alternating pattern of increases and decreases, which was evident by the previously described negative correlations in the year dimension for the day/year interaction (Table 5).

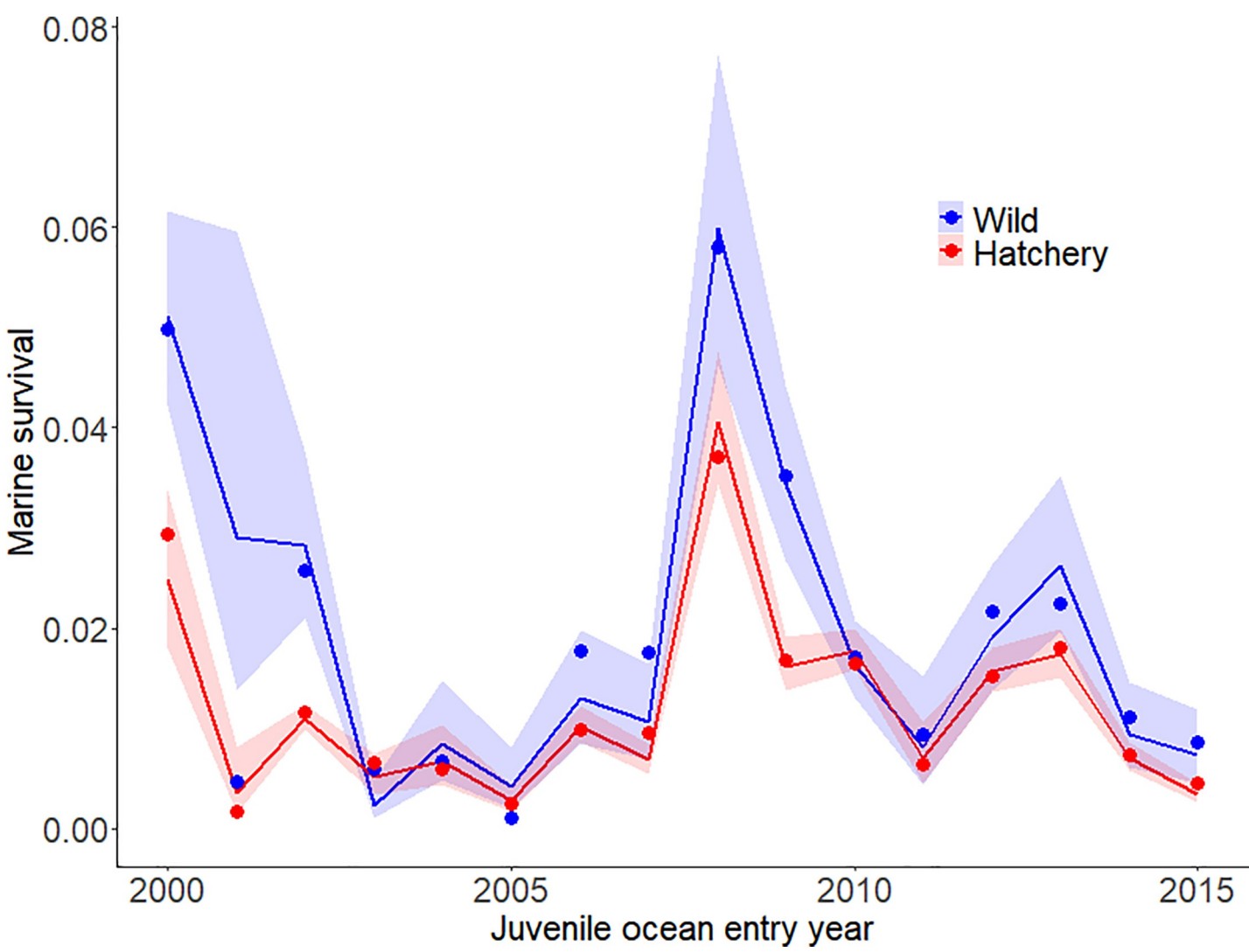

**Fig 5. Annual survival of spring/summer Chinook salmon in the Columbia River.** Observed (points) and estimated (line) annual survival with 95% confidence intervals (polygons) for hatchery (red) and wild (blue) spring/summer Chinook salmon from 2000 through 2015.

## Model validation

The AUC statistics for the hatchery and wild fish models were equal to 0.69 and 0.76, respectively. This indicates that the ability for the hatchery fish model to discriminate between true positives and false positives was slightly below the acceptable threshold of 0.7, while the wild fish model was above it. Visual inspection of the simulation experiments suggests that the estimation model provided unbiased estimates (i.e., the center of mass of the violins is near zero) for the fixed effects in the TMB model (S1 Fig). Across all experiments, some random draws led to negative biases for the standard deviations of the day effect ($\psi$). These biases were usually associated with random draws with low numbers of surviving fish and little auto-correlation in the day effect. As sample sizes increased, the precision increased (i.e., the violins get vertically compressed) for the correlations and standard deviations that describe the random processes for day and day/year. Experiments examining the magnitude of the correlation for both the day ($\tau$) and day/year ($\rho$ and $\pi$) effects resulted in no bias in the other fixed effects (i.e.,

average deviation was zero). Additionally, as the magnitude of the correlations for the random processes increased, the precision increased for the estimated correlations, decreased for the mean survival ($\mu$), and remained unchanged for the marine covariates ($\beta_{PDO.sum}$ and $\beta_{CUI.spr}$) where the subscripts refer to specific covariate names in Table 2.

When we compared the performance of the generalized linear model (glm) with fixed-effects for day, day$^2$, and day/year interaction using the glm function in R with the random effects model in TMB, we found little difference in the bias and precision of the mean survival ($\mu$) and the marine covariates ($\beta_{PDO.sum}$ and $\beta_{CUI.spr}$) (Fig 6, upper panel). Examining a single realization for the simulated data, the survival estimates from glm model (S2 Fig, red line) and TMB model (blue line) were similar. However, the random processes in the TMB model provided more flexibility to capture the daily variability in survival within a year. While there were almost no differences in the bias or precision for the mean survival and environmental covariates, the standard errors for those fixed-effects in the TMB model were between 65%-70% higher than the glm model (Fig 6, lower panel).

## Discussion

We found that for Snake River spring/summer Chinook salmon, survival in the ocean was strongly related to several indicators of ocean conditions and arrival timing in the estuary. Arrival timing is the culmination of processes that occur in the freshwater, so we have established strong linkages between freshwater conditions and ocean conditions. Our modeling framework allows for historical trends but also has the flexibility to forecast trends into the future. Perhaps counterintuitively, increasing the flexibility of the model and allowing more of the uncertainty to be explained by these temporal processes led to increased uncertainty in the mean survival and environmental covariates (Fig 6). Thus, this research highlights that conclusions about the uncertainty in the survival estimates must also reflect the uncertainty in the processes that are believed to affect survival (i.e., timing). Additionally, our generalized approach for integrating random variability into an SAR model can easily be expanded from the AR1 "lattice" for the day/year interaction to higher dimensional interactions that include biological forces such as size and weight, or environmental forces such stream temperature. Because these forces are associated with "levers" that managers of freshwater systems can manipulate–as opposed to climate conditions, quantifying the effect of these interactions on the uncertainty in survival is critically important for evaluating future management scenarios.

### Hatchery-wild comparisons

Different rearing types of spring/summer Chinook salmon exhibit differences in SAR within and between years. There are expected differences between fish reared in a hatchery and fish exposed to natural conditions in the wild, including size, condition, risk aversion, arrival timing, parasite load, and numerous other factors. We clearly documented the effect of arrival timing on marine survival was not consistent between fish of different rearing types, and we described two primary differences in timing and marine survival. First, the arrival timing distribution for juvenile salmon differs between the hatchery and wild fish, with hatchery fish clumped in a narrow window, mostly completed by early June. In contrast, wild fish start to arrive earlier and the distribution has a long tail, with some fish not passing Bonneville Dam until mid-July. Second, on average across years, survival peaks for wild fish migrating early and then declines throughout the remainder of the migration, whereas hatchery fish, on average, show no consistent pattern in survival across years based on arrival timing.

There are multiple reasons why wild fish may be more sensitive to arrival timing than hatchery fish, though much of this is speculation. Perhaps the most likely cause is the

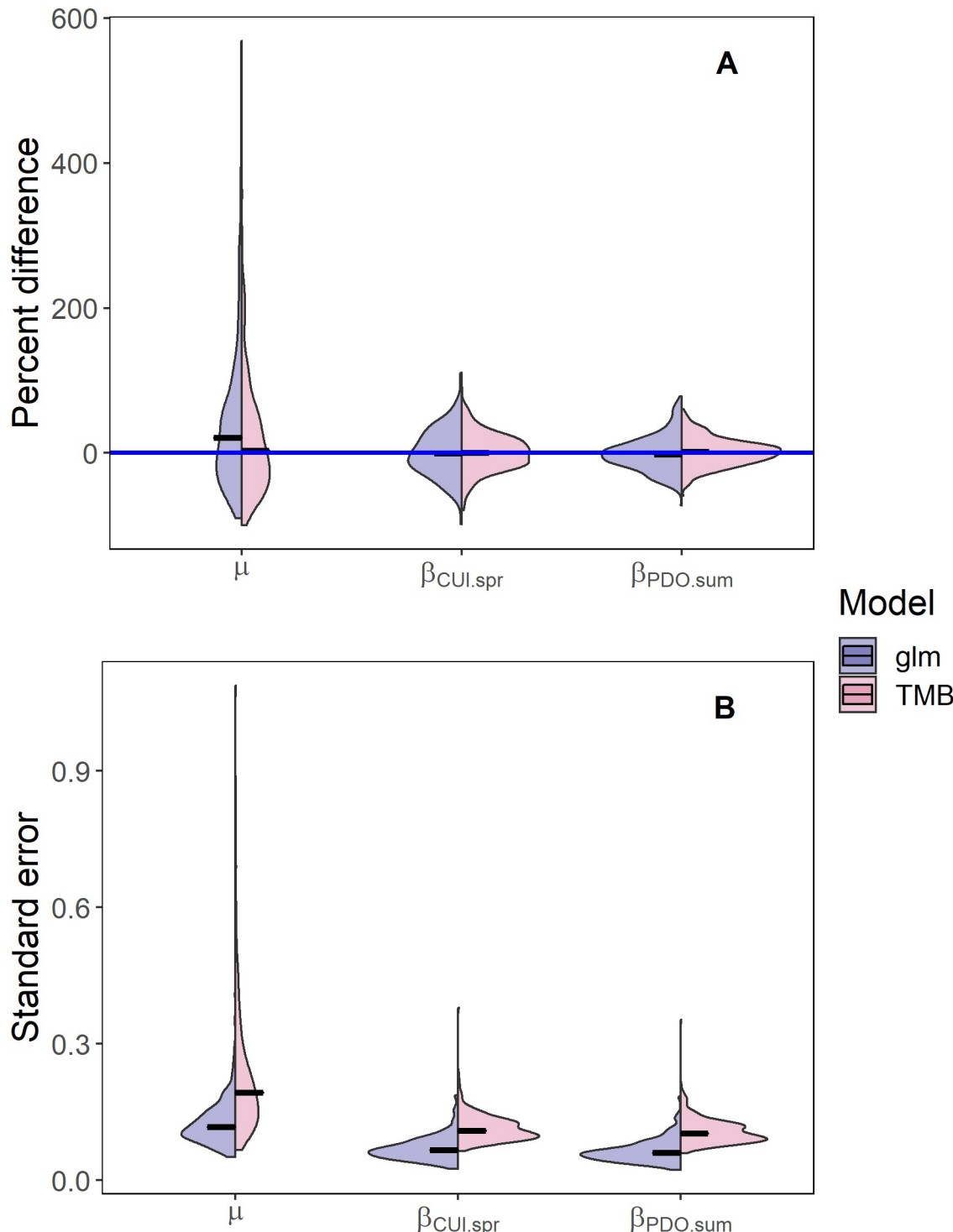

**Fig 6. Differences between the mean survival and standard error of the mean survival for models with fixed and random effects.** Split violin plot comparing the percent difference (upper panel) between the estimated and true parameter values and the standard errors for the fixed effects $\mu$, $\beta_{CUI.spr}$, and $\beta_{PDO.sum}$ (lower panel) for the mixed-effect model with AR1 processes for day and day/year interaction (TMB; blue violins) and the fixed-effect model for day, day2, and day/year interaction (glm; red violins) fit to simulated data for wild spring/summer Chinook salmon. Simulated the data were generated from the model with the lowest AIC for wild fish (see Table 4). Horizontal lines represent median values for the violins and the horizontal blue line in panel (A) represents zero percent difference between the estimated and true parameter values.

difference in size between the two groups. If early marine survival is size-dependent, which has been shown for other salmon stocks [13,27–29], the larger size of hatchery fish could afford them some level of independence from predators. Additionally, large subsidies of hatchery smolts may increase the density of the predator communities, and these predators may differentially select for wild fish because they are smaller and more available once the pulse of hatchery fish has passed [12].

## Arrival timing

A key component of this model is the inclusion of arrival timing to the marine environment. Gosselin et al. [7] showed that management practices in freshwater can have large impacts on marine survival via carryover effects, which can materialize in the form of altered fish size or timing at out-migration. Although size-dependent mortality is important, we focused on the impacts of timing for this effort. Arrival timing has been shown to be an important catalyst for carryover effects [7] and these data are quite readily available, as each fish detected at Bonneville Dam has its own time stamp. However, there is a large amount of variability in arrival timing, and managers of wild salmon populations have few levers to manipulate the environmental experiences that may influence marine survival. To the extent that the freshwater environment influences salmon behavior, performance, growth, and survival in the marine environment, these influences should be incorporated into modeling efforts aimed at understanding salmon marine ecology. Freshwater conditions affecting arrival timing (e.g., flow and temperature) are likely to be correlated with conditions in the marine environment [30], and phenological variability in the marine ecosystem is driven by atmospheric and oceanographic processes with substantial inter-annual variability [31]. For example, wind-driven ocean currents transition from south to north each spring, initiating a strong upwelling of deeper ocean water. The nutrients in this upwelled water can spawn or feed a spring phytoplankton bloom [32]. Moreover, the newly transitioned currents can bring species of zooplankton such as copepods that are high in fatty acids [33,34], further enriching the production at lower trophic levels. Salmon eventually benefit from these dynamics, but the timing and magnitude of local production varies from year to year. Although salmon have evolved to optimize arrival timing on average [35], the broad distribution of arrival timing may be a bet-hedging strategy [36,37] to ensure some fish arrive at the ocean when conditions are optimal. If future freshwater management practices significantly alter the mean arrival timing or the variability in timing, this could have important, and perhaps unforeseen, effects on marine survival. Similarly, if climate changes in either the freshwater or marine environment result in a mismatch between salmon arrival timing and optimal arrival timing, marine survival will be impacted. These interactions are a clear demonstration of the importance of carryover effects and a direct link between salmon survival and management decisions that may affect arrival timing [38].

## Marine covariates

The top performing models describing Chinook salmon marine survival included three categories of environmental covariates for wild fish (i.e., basin-scale sea surface temperature ('ersstArc' and PDO), a local sea surface temperature ('ersstWA'), and a regional spring upwelling variable ('cui')), and three categories of environmental covariates for hatchery fish (i.e., a measure of alongshore flow ('transport'), ocean circulation ('NPGO'), and sea surface temperature ('errstArc')). For each rearing type, there are logical links between the metric and multiple oceanographic or ecosystem processes that could influence salmon growth and survival. However, most of these links are indirect and rely on other oceanographic factors. For example, local sea surface temperature can influence growth rates directly [39], but a more likely

influence on salmon survival involves production at lower trophic levels and temperature-dependent distribution of prey and predator species [40].

In this effort, we intentionally restricted our potential ocean covariates to publicly-available (and mostly physical) variables. These variables do not necessarily directly relate to the ecosystem processes that determine salmon survival, but rather represent correlations with these processes. Some biological time series that more directly characterize ecosystem processes such as trophic dynamics are available, but only for recent years (e.g., stoplight chart for ocean survival estimates, http://www.nwfsc.noaa.gov/oceanconditions). For other research goals, such as near-term forecasting, these more direct metrics may be more appropriate. Indeed, as more biological data are collected, reliance on correlations should be reduced [41] and the use of mechanistic ecosystem models will become more important [42,43].

## Model fit

Comparing the residual deviance ratio, defined as the fit of a particular model relative to the model where each data point has its own parameter, the fixed effects models that included only marine covariates had ratios equal to 0.077 and 0.167 for wild and hatchery fish, respectively. When we removed the marine covariates and included a day/year interaction, the ratios increased to 0.197 and 0.346, respectively. Finally, the ratios increased to 0.208 and 0.350, respectively, for the model that included marine covariates and random effects for day and the day/year interaction (Table 6). The small differences in the ratios between the random effects models with and without marine covariates does not imply that marine conditions do not affect Chinook salmon survival. In fact, as shown by the estimated magnitude of the deviates in Fig 1, the marine covariates were correlated with large differences in marine survival. However, rather than a uniform response of all fish to the marine conditions in a particular year, our model demonstrates that the timing of when the juvenile salmon encounter the marine conditions appears to explain more of the data (Table 6). The mechanism that is driving this differential survival across days and years remains a critical knowledge gap and a focus of future salmon modeling.

Combining impacts from multiple environments has been applied in several past efforts to model Snake River spring/summer Chinook marine survival. The day effect was described by Scheuerell et al. [6] using a quadratic effect for day in a logistic regression model and showed that earlier fish tend to have higher survival, but this shifted somewhat from year to year. Holsman et al. [44] also use a logistic regression for this ESU and characterized the impacts of predators, prey, flow, and the temperature difference between the Columbia River and the nearshore ocean; however, they did not include a day effect in their model. Similarly, Haseker

**Table 6. Deviance ratios for top models.**

| Model | Likelihood | | Deviance ratio | |
|---|---|---|---|---|
| | Hatchery | Wild | Hatchery | Wild |
| Null | 1760.4 | 998.3 | 0 | 0 |
| Marine | 1475.2 | 921.6 | 0.162 | 0.077 |
| Day | 1718.6 | 952.1 | 0.024 | 0.047 |
| Day/year | 1151.4 | 802.4 | 0.346 | 0.197 |
| Day + day/year | 1151.4 | 800.4 | 0.346 | 0.198 |
| Marine covariates + day + day/year | 1144.1 | 790.8 | 0.350 | 0.208 |

Deviance ratios (i.e.,$1-D_m/D_0$) for different fixed and mixed-effects models for hatchery and wild fish, where $D_m$ is -2 times the log-likelihood of model m and $D_0$ is -2 times the log-likelihood of the null model. The deviance ratio explains how close the model is to a model that fits the data exactly.

et al. [45] demonstrated the importance of river flow (the proportion of water spilled over dams and migration rate), in modeling marine survival for this ESU, but included a linear effect of day. Miller et al. [46] used a logistic regression to show that the size at out-migration was not as important as the size at marine capture (after fish had been in the ocean for weeks to months), suggesting that marine growth is highly influential in setting mortality rates. Finally, Gosselin et al. [7] used a mixed effects regression to describe carryover effects from the freshwater environment, with particular emphasis on transportation impacts on hatchery and wild fish, but constrained the underlying process for the day effect to be quadratic. Our current model design represents a compromise between model complexity, realism, and the clear need to address the interactions between freshwater impacts and the marine ecosystem. Rather than treating the effect of timing on survival as a fixed effect described by a linear or quadratic relationship, our model accounts for the heterogeneity in the survival processes by treating the effect of timing as random process.

We recognize that there are multiple ways to evaluate model fit and specification (i.e., fixed and random effects structures) for mixed effects models. For instance, Vaida and Blanchard [47] propose using conditional AIC, because marginal AIC tends to favor smaller models with fewer random effects [48]; however, conditional AIC is not computationally reasonable for large data sets such as ours [48]. Zuur [49] proposed, in the case of REML models (restricted maximum likelihood), selecting the number random effects using marginal AIC conditioned on all of the fixed effects, and then choosing the fixed effects conditional on the structure of the optimal random effects. Because we have not implemented REML in our joint likelihood, and given that we have greater than 1600 observations [48], we have chosen to compare models using the marginal AIC and recognize that we may be underestimating the optimal number of random effects. Future analyses may also consider using conditional AIC approaches (e.g., the DHARMa package in R [50]) to evaluate model misspecification for a larger suite of random effects models; however, in this paper we have specifically focused on evaluating the effects of the unknown processes for day and year.

## Using the model prospectively

We have demonstrated that our model is powerful for detecting effects in both marine and freshwater environments from historical data. However, we designed the model such that it can also be used for population viability modeling. To do this, the ocean survival model is incorporated into a stochastic age-based life cycle model (e.g., Zabel et al. 2006). This approach has been adopted by NOAA Fisheries to examine the effects of climate and climate change on salmon population viability [51]. The fact that several of the most important ocean indicators (e.g., SST) are amenable to forecasting under climate change scenarios allows for an important examination of how Snake River spring/summer will respond to future climate conditions.

## Caveats

We included arrival timing, but did not include other attributes such as fish size, which is known to have important impacts on trophic interactions and size-dependent survival [13,28,29,52]. Miller et al. [46] showed that Snake River spring/summer Chinook marine survival was more related to size after some period of ocean residence than size at out-migration, but did not rule out the possibility that some level of size-dependent mortality did not already occur. We note that although fish size is known to affect migration pathways through the hydrosystem [53], detections at Bonneville Dam did not show a significant size bias (see Fig 3 in Faulkner et al.). Nonetheless, we acknowledge that there may be other ways in which detected fish were not fully representative of the run-at-large, so our conclusions apply

specifically to this set of fish. Further research to extend this model is necessary to fully understand how the interaction of other fish attributes such as size in the freshwater environment are likely to affect marine survival. Fortunately, given the flexibility of the multivariate framework, such analyses are possible with the availability of additional data. Additionally, maturation schedules, the fraction of a salmon maturing and returning to spawn at different ages, are also size-dependent–larger and faster growing fish tend to mature earlier [2]. Recent spawner-recruit analysis suggests that climate conditions affect both the maturation schedule and the survival of some stocks of salmon [54]; however, timing and size were not a part of these models. Future iterations of our model could examine the effects of size and maturation simultaneously, with the goal of understanding how management actions in freshwater environment affect size, maturation, and ultimately, survival.

We view our model as a robust approach for integrating the freshwater and marine effects in a single estimation model. By partitioning the different sources of uncertainty between the observation model (binomial likelihood) and process models (random effects for day, year, and day/year interactions) we provide a more accurate estimate of the uncertainty and relative importance of the fixed effects associated with the marine covariates relative to the random deviations in survival associated with differences in arrival timing between years. While our model was restricted to examining the two-dimensional interaction between day and year, this model can quickly be scaled-up to higher-dimensional questions related to the interaction between day, year, size, and maturation.

## Supporting information

**S1 Table. R scripts.** All files and model output are available for upload from the github.com/bchasco/sar_paper.
(DOCX)

**S1 Fig. Estimates of parameter bias.** Violin plot of the percent difference between the estimated and "true" parameter values (rows) for three experiments (columns) related to sample size ($n_{jt}$), correlation of the daily random effects ($\rho_j$), and correlation of the day/year random effects ($\tau^{(j)}$ and $\tau^{(t)}$). The simulated data for the wild spring/summer Chinook salmon is based on the vectors of maximum likelihood parameters estimates ($\theta_{mle}$ and $\gamma_{mle}$, yellow violins), or the manipulation the sample size or some element of those vectors based on different trials (h; x-axis) and experiment (e; columns). For compactness, we removed the r subscript and superscript for the parameters since all simulations are for wild fish. To recreate the results of these simulation experiments refer to the S1 Table.
(TIF)

**S2 Fig. Simulated data and model fit for a simulation realization.** A single realization of the simulated smolt-to-adult (SAR; grey points) for wild spring/summr Chinook salmon based on the mle estimates for the simulation model with AR1 processes for the day and day/year interactions. The blue lines represent the SAR estimates for TMB estimation model with AR1 process for day and day/year, and the red lines represent the glm model implemented in R with fixed-effects for day, day$^2$, and the day/year interaction.
(TIF)

## Acknowledgments

We would like to thank Jeff Jorgenson and David Huff for their reviews of previous versions of this manuscript, Susan Iltis for helping us to compile PIT tag data for this analysis, and Jennifer Gosselin for listening to our initial formulations of the model.

## Author Contributions

**Conceptualization:** Brandon Chasco, Brian Burke, Lisa Crozier, Rich Zabel.

**Formal analysis:** Brandon Chasco, Brian Burke.

**Investigation:** Brandon Chasco.

**Methodology:** Brandon Chasco.

**Project administration:** Rich Zabel.

**Software:** Brandon Chasco.

**Supervision:** Brian Burke.

**Validation:** Brandon Chasco.

**Writing – original draft:** Brandon Chasco, Brian Burke, Lisa Crozier, Rich Zabel.

**Writing – review & editing:** Brandon Chasco, Brian Burke, Lisa Crozier, Rich Zabel.

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
