## [Decision Letter · Decision Letter 0]

17 Jul 2020

PONE-D-20-13616

Differential impacts of freshwater and marine covariates on wild and hatchery Chinook salmon marine survival

PLOS ONE

Dear Dr. Chasco,

Thank you for submitting your manuscript to PLOS ONE. After careful consideration, we feel that it has merit but does not fully meet PLOS ONE’s publication criteria as it currently stands. Therefore, we invite you to submit a revised version of the manuscript that addresses the points raised during the review process.

I found this to be an interesting and well written manuscript and the reviewer agreed. Unfortunately, the second reviewer failed to return a review for the manuscript, so in the interest of returning a prompt decision to the authors I will act as the other reviewer. The reviewer felt this manuscript only needs minor revision and I agree. The reviewer has provided comments to assist the authors in their review and I have provided some editorial comment as there are some formatting and grammatical errors in the manuscript.

We look forward to receiving your revised manuscript.

Kind regards,

Heather M. Patterson, Ph.D.

Academic Editor

PLOS ONE

Journal Requirements:

2.Thank you for stating the following financial disclosure:

 [NO].

3.Thank you for stating the following in your Competing Interests section: 

[NO].

Reviewers' comments:

Reviewer's Responses to Questions

**Comments to the Author**

1. Is the manuscript technically sound, and do the data support the conclusions?

Reviewer #1: Yes

2. Has the statistical analysis been performed appropriately and rigorously? 

Reviewer #1: Yes

3. Have the authors made all data underlying the findings in their manuscript fully available?

Reviewer #1: Yes

4. Is the manuscript presented in an intelligible fashion and written in standard English?

Reviewer #1: Yes

5. Review Comments to the Author

Reviewer #1: Review of Chasco et al. PONE-D-20-13616

Chasco et al. use a logistic regression model with random effects for year and day of year to quantify the effect of marine, environmental, and freshwater (e.g., river flow) covariates on smolt-to-adult return rate of spring-summer Chinook salmon in the Columbia River. The chief contribution of this manuscript is the inclusion of (autocorrelated) random effects for day of year, which allows SAR to vary with day of passage at Bonneville Dam in an unstructured manner such that the data itself inform how SAR varies over days within years and among years. Previous approaches have included day of year but have imposed a specific structure on the form of temporal variability (i.e., linear or quadratic fixed effects). In contrast, the approach used by the authors makes no assumptions about how SAR varies with day of year, allowing the authors to more fully explore the nature of within and among year variation in SAR. Toward this end, while the manuscript draws inference on the important effect of annual-level covariates on SAR, it devotes equal (if not more) attention to exploration of random effects within and among years and among rearing types. The manuscript is well written and the model well developed and fully analyzed (e.g., simulation analysis, comparison with glm models). For these reasons, I find the manuscript acceptable for publication with attention to some minor comments.

Minor comments:

“carryover effects” – The introduction brings in the idea of carryover effects, but the link between carryover effects and arrival timing in the estuary is weakly developed in the introduction. That is, at first glance, arrival timing itself does not seem to be a carryover effect from freshwater to the estuary/ocean. Size at outmigration, if size is mediated by the freshwater environment, is an example of a direct measure of a carryover effect if size at outmigration subsequently affects SAR. The argument for arrival timing as a carryover effect is better supported in the discussion. However, because carryover effects are such an important concept in this paper, I think it would be worth spending a bit more time in the introduction to explain to the reader why arrival timing can be considered as a carryover effect. If you can get across the idea that the freshwater environment itself (and management thereof) can influence arrival timing, and arrival timing affects SAR, then it better establishes the idea of arrival timing is a carryover effect and will better set up the rest of paper.

The analysis is conditioned on fish detected at Bonneville Dam. So what are the consequences of ignoring fish that survived to Bonneville but were not detected? One thought is that “annual” survival should be interpreted with caution, as this is the weighted average survival of fish detected at Bonneville. Although unbiased with respect to this strict definition (fish detected at Bonneville), it could be a biased estimate for all PIT-tagged spring/summer Chinook, particularly if detection probability varies with day of year (which is likely). Second, Faulkner et al. (2019; https://doi.org/10.1002/tafs.10200) recently found that detection probability at Columbia River dams increased with fish size, suggesting that the analysis by Chasco et al. uses fish that are likely to be larger than the run-at-large passing Bonneville Dam. Conditioning on detected fish vastly simplifies modeling, and although I don’t think this is a major issue, the authors should at least explicitly acknowledge that 1) many PIT-tagged fish passing Bonneville are not included in the analysis, and 2) the potential consequences of such exclusion.

Line 85: “barged downstream as juveniles”. Explain more fully otherwise readers outside of the Columbia Basin will have no idea what this means.

Lines 85-86: Why exclude fish migrating prior to April 9th and after July 8th? The fact the that they comprise <0.14% of the total observations is just as much an argument for including these observations as excluding them.

6. PLOS authors have the option to publish the peer review history of their article (what does this mean?). If published, this will include your full peer review and any attached files.

Reviewer #1: No

---

## [Author Response · Author response to Decision Letter 0]

27 Aug 2020

Minor comments:

“carryover effects” – The introduction brings in the idea of carryover effects, but the link between carryover effects and arrival timing in the estuary is weakly developed in the introduction. That is, at first glance, arrival timing itself does not seem to be a carryover effect from freshwater to the estuary/ocean. Size at outmigration, if size is mediated by the freshwater environment, is an example of a direct measure of a carryover effect if size at outmigration subsequently affects SAR. The argument for arrival timing as a carryover effect is better supported in the discussion. However, because carryover effects are such an important concept in this paper, I think it would be worth spending a bit more time in the introduction to explain to the reader why arrival timing can be considered as a carryover effect. If you can get across the idea that the freshwater environment itself (and management thereof) can influence arrival timing, and arrival timing affects SAR, then it better establishes the idea of arrival timing is a carryover effect and will better set up the rest of paper.

We agree that migration timing should feature more prominently and have included the following addition to the fourth introductory paragraph.

'Carryover effects are effects that “carry over” from one life stage to another [17]. We propose the working definition: in an ecological context, carryover effects occur in any situation in which an individual’s previous history and experience influences their current performance in a given situation. While effects such as length, weight, and freshwater environmental covariates may be explored in future analyses [7], we focused our attention on how phenology (i.e., migration timing) in the freshwater environment (the previous history) carries over to the survival in the marine environment (the current life stage). In part, we have focused our attention on phenology because of the large emphasis that co-managers of the Columbia Basin have placed on restoring the natural migration conditions for juvenile salmon. In particular, the practice of transporting fish in barges around dams and spilling more water over the top of dams has decreased delay, resulting in fish arriving at the estuary earlier.'

The analysis is conditioned on fish detected at Bonneville Dam. So what are the consequences of ignoring fish that survived to Bonneville but were not detected? One thought is that “annual” survival should be interpreted with caution, as this is the weighted average survival of fish detected at Bonneville. Although unbiased with respect to this strict definition (fish detected at Bonneville), it could be a biased estimate for all PIT-tagged spring/summer Chinook, particularly if detection probability varies with day of year (which is likely). Second, Faulkner et al. (2019; https://doi.org/10.1002/tafs.10200) recently found that detection probability at Columbia River dams increased with fish size, suggesting that the analysis by Chasco et al. uses fish that are likely to be larger than the run-at-large passing Bonneville Dam. Conditioning on detected fish vastly simplifies modeling, and although I don’t think this is a major issue, the authors should at least explicitly acknowledge that 1) many PIT-tagged fish passing Bonneville are not included in the analysis, and 2) the potential consequences of such exclusion.

We agree with the reviewer on this point as well and thank them for directing us to the Faulkner paper. We have added the following text in the discussion.

'We note that although fish size is known to affect migration pathways through the hydrosystem (Faulkner et al. 2019), detections at Bonneville Dam did not show a significant size bias (see Fig 3 in Faulkner et al 2019). Nonetheless, we acknowledge that there may be other ways in which detected fish were not fully representative of the run-at-large, so our conclusions apply specifically to this set of fish. Further research to extend this model is necessary to fully understand how the interaction of other fish attributes such as size in the freshwater environment are likely to affect marine survival.'

Line 85: “barged downstream as juveniles”. Explain more fully otherwise readers outside of the Columbia Basin will have no idea what this means.

We have added,

Lines 77 – 79, 'In particular, the practice of transporting fish in barges around dams and spilling more water over the top of dams has decreased delay, resulting in fish arriving at the estuary earlier'. And Lines 108 to 110, 'and fish that did not volitionally migrate (i.e., placed into barges to avoid passage through the hydrosystem) downstream as juveniles'

Lines 85-86: Why exclude fish migrating prior to April 9th and after July 8th? The fact the that they comprise <0.14% of the total observations is just as much an argument for including these observations as excluding them.

We added the following,

'There is very little data to inform the temporal autocorrelation at the margins of the migration period, and an initial analysis demonstrated that removing these observations greatly improved the speed and convergence of the model fit with little change in the estimates of model parameters.'

---

## [Decision Letter · Decision Letter 1]

13 Nov 2020

PONE-D-20-13616R1

Differential impacts of freshwater and marine covariates on wild and hatchery Chinook salmon marine survival

PLOS ONE

Dear Dr. Chasco,

Thank you for submitting your manuscript to PLOS ONE. After careful consideration, we feel that it has merit but does not fully meet PLOS ONE’s publication criteria as it currently stands. Therefore, we invite you to submit a revised version of the manuscript that addresses the points raised during the review process.

I have recently taken over the editing responsibilities for this paper and requested an additional review to meet the minimum suggestion of 2 reviewers. My apologies for the delay this has caused. This second review has highlighted some issues that resulted in the recommendation for “Major revisions” prior to publication.

In particular, please respond to Reviewer 2’s comment #6 on the use of AIC to compare mixed-effects models. Below I have listed a few additional points (and some minor edits attached) to consider:

I recommend following Edwards and Augre-Mth (2018) guidance on scientific notation. For example, they suggest avoiding labels in superscripts if possible to avoid confusion, as in equation 7 (is D an exponent or a label?), using italics when using English letters are used for variables, using lower case bold for vectors and upper case bold for matrices, etc. Also, I suggest using lower-case sigma^2 to represent variance terms, with labels in the subscript to differentiate variance terms (and upper case Sigma for a covariance matrix, as you’ve done). Also, given that rearing types (hatchery and wild) were modelled independently, you could keep the *r* subscript out of the equations entirely and just mention that you ran the model on two data sets: wild and hatchery. This would simply the equations significantly.   [Edwards, AM, Auger‐Méthé, M. Some guidance on using mathematical notation in ecology. Methods Ecol Evol. 2019; 10: 92– 99. https://doi.org/10.1111/2041-210X.13105]How were the environmental variables in Table 2 chosen? Some justification for their choice would be valuable.Text justifying the inclusion of random effects would also be useful. For example, in the introduction, I suggest a more complete description of why day and year variables were treated as random effects, given this is a key advance over previous studies (as described in the 1^st^ paragraph). I suggest moving the justification for work on page 15 (top of paragraph 2) to the Introduction. Also, why might the day/year interaction term be biologically meaningful, justifying its inclusion? If previous studies did not include autocorrelation in day and year terms, why did you consider this a shortcoming, justifying its inclusion here (is there evidence for autocorrelation in those factors, or autocorrelation in the underlying biological mechanisms for the day and year effects)? This is perhaps especially important given that you found including all random effects resulted in model over-parameterization.

We look forward to receiving your revised manuscript.

Kind regards,

Carrie A. Holt

Academic Editor

PLOS ONE

Reviewers' comments:

Reviewer's Responses to Questions

**Comments to the Author**

1. If the authors have adequately addressed your comments raised in a previous round of review and you feel that this manuscript is now acceptable for publication, you may indicate that here to bypass the “Comments to the Author” section, enter your conflict of interest statement in the “Confidential to Editor” section, and submit your "Accept" recommendation.

Reviewer #1: All comments have been addressed

Reviewer #2: (No Response)

2. Is the manuscript technically sound, and do the data support the conclusions?

Reviewer #1: Yes

Reviewer #2: Yes

3. Has the statistical analysis been performed appropriately and rigorously? 

Reviewer #1: Yes

Reviewer #2: No

4. Have the authors made all data underlying the findings in their manuscript fully available?

Reviewer #1: Yes

Reviewer #2: Yes

5. Is the manuscript presented in an intelligible fashion and written in standard English?

Reviewer #1: Yes

Reviewer #2: Yes

6. Review Comments to the Author

Reviewer #1: (No Response)

Reviewer #2: The authors have satisfactorily addressed the comments of the original review by another Reviewer. One of the comments of that Reviewer was that about the need to explain how migration timing can be considered a carryover effect. While the authors did a good job in expanding that explanation, I feel that one element is missing. The authors mention that they focused on migration timing as a carry over effect because it is influenced by management practices in freshwater (i.e. transporting fish, spilling more water). However, not all salmon bearing systems are heavily managed like the Columbia. Migration timing in most other systems will be driven by environmental conditions experienced in freshwater and I suggest the authors also explain that in the paragraph. That would help broadening the application of the carry over ideas presented in the paper.

I also have a few more comments, specifically related to data analysis that should be addressed before publication:

1) The term multivariate seems to be used in the text to refer to a model with multiple predictor variables. Strictly speaking, multivariate refers to a model with more than one response variable. However, the authors only have one response variable in their model (SAR), so the term multivariate should not be used.

2) Page 9, equation 2 and associated text. If beta is a vector, please present in in bold. The term x_t should be matrix (where rows are measurements and columns are the different predictors), not a vector as described in the text. The text should be corrected and the term should be represented as a bold capital X. Correction should be applied to Table 3 as well.

3) Page 11, first paragraph. The text implies that SAR (s_rjt) is binomially distributed. Actually, s_rjt is a parameter of the binomial distribution. What is binomially distributed is the number of survivors (k_rjt). Please revise.

4) The authors frequently refer to the top model(s) as the models that had best-fit. AIC is not simply a measure of fit but rather it is a statistics that represents a balance between model fit and complexity. Therefore, it's not appropriate to refer to the selected model(s) as the model(s) of best-fit (a model with best fit may not even be included among the selected models). I suggest referring to selected models as more parsimonious or optimal model(s).

5) Page 16. The authors refer to the intervals as credible intervals. The term credible is used for interval determined using Bayesian inference, but it seems the authors are using a frequentist inference framework. So, to avoid confusion I suggest they refer to the interval as confidence interval throughout the paper.

6) To my knowledge, marginal AIC should not be used to conduct model selection in linear mixed models in the way it was conducted by the authors (i.e. comparing models with different fixed and random effects structures). I suggest the authors check out the paper by Vaida & Blanchard (2005) and the book by Zuur et al. (2009). In the latter, the authors suggest to first select the optimal random effects structure by applying AIC to models with all fixed effects but different random effects. Once the optimal random structure is selected, AIC can then be used to compare models that have different fixed effects but the same (previously selected) optimal random effects structure.

Vaida F, Blanchard S. Conditional Akaike information for mixed-effects models. Biometrika. 2005;92:351–70.

Zuur AF, Ieno EN, Walker NJ, Saveliev AA, Smith GM (2009) Mixed effects models and

extensions in ecology with R. New York: Springer. 574 p.

7) It seems Figs. 4 and 5 were uploaded in the wrong order. Please check.

7. PLOS authors have the option to publish the peer review history of their article (what does this mean?). If published, this will include your full peer review and any attached files.

Reviewer #1: No

Reviewer #2: No

---

## [Author Response · Author response to Decision Letter 1]

21 Dec 2020

Editor’s comments:

 I recommend following Edwards and Augre-Mth (2018) guidance on scientific notation. For example, they suggest avoiding labels in superscripts if possible to avoid confusion, as in equation 7 (is D an exponent or a label?), using italics when using English letters are used for variables, using lower case bold for vectors and upper case bold for matrices, etc. Also, I suggest using lower-case sigma^2 to represent variance terms, with labels in the subscript to differentiate variance terms (and upper case Sigma for a covariance matrix, as you’ve done). Also, given that rearing types (hatchery and wild) were modelled independently, you could keep the r subscript out of the equations entirely and just mention that you ran the model on two data sets: wild and hatchery. This would simply the equations significantly. [Edwards, AM, Auger‐Méthé, M. Some guidance on using mathematical notation in ecology. Methods Ecol Evol. 2019; 10: 92– 99. https://doi.org/10.1111/2041-210X.13105]

We have updated the equations in accordance with the format suggested by Edwards and Auger-Methe.

 How were the environmental variables in Table 2 chosen? Some justification for their choice would be valuable.

We have included the following text on lines 141 to 151

Variables represent large-scale oceanographic patterns as well as regional physical metrics. While not all variables have a proposed direct mechanistic relationship with salmon survival, they have been shown to correlate with Chinook salmon returning to specific ESUs [20,21]. The environmental variables in our models include basin-scale oceanic and atmospheric variables (i.e., North Pacific Gyre Oscillation (PGO), Oceanic Niño Index (ONI), Multivariate ENSO Index (MEI), and North Pacific Index (NPI)), local or regional variables (i.e., sea surface temperature for coastal Washington (ersstWACoast), sea surface temperature from Johnstone and Mantua (2014) (ersstARC), coastal upwelling index (CUI), and measure of Sverdrup index most correlated with the temperatures in the upper 20 meters (transport)), and indicators from the Columbia River representing the environment that salmon inhabited just prior to migrating into the ocean (i.e., Columbia River flow (CRflow) and Columbia River temperature (CRtemp)).

 Text justifying the inclusion of random effects would also be useful. For example, in the introduction, I suggest a more complete description of why day and year variables were treated as random effects, given this is a key advance over previous studies (as described in the 1st paragraph). I suggest moving the justification for work on page 15 (top of paragraph 2) to the Introduction. Also, why might the day/year interaction term be biologically meaningful, justifying its inclusion? If previous studies did not include autocorrelation in day and year terms, why did you consider this a shortcoming, justifying its inclusion here (is there evidence for autocorrelation in those factors, or autocorrelation in the underlying biological mechanisms for the day and year effects)? This is perhaps especially important given that you found including all random effects resulted in model over-parameterization.

We included the following text in lines 79-101. Some of the text currently in lines 89-105 had previously been included in the discussion – as per the editors comments.

Previous estimates of smolt-to-adult returns (SAR) used either generalized linear models (glm) and treated the temporal variability in survival with fixed effects for day, day2, and the day/year interaction [6], or generalized mixed effects models (glmm) with day and or year as independent uncorrelated random effects [7]. Numerous research efforts have shown that not accounting for autocorrelation in fisheries data can lead to biases in parameters estimates and derived variables of the models [19], and it is unknown whether the fixed effects for day and day/year interactions can lead to biases in the expectation and uncertainty in salmon survival models. 

In this effort, we provide a generalized statistical model that scientists and managers can use to integrate the complex interacting effects of environmental conditions across multiple habitats with estimates of salmon survival. From a modeling perspective, our justification for including autocorrelated random effects for the day, year, and day/year interactions recognizes that natural systems have inherent variability that a fixed effect model is unlikely to capture, and a random effects model allow us to decouple the uncertainty in the day effect from the uncertainty in the observations. From a biological perspective, other factors not measured during the migration period are also likely to affect salmon survival (e.g., predation) and therefore the effect of migration day may not be as smooth as a quadratic curve that is constant among years. Our model is meant to provide a parametric estimate of the autocorrelation at the daily and annual time scales, recognizing that these data were collected over time and the independence between observations is likely to be affected by the duration between observations. To evaluate our approach relative to previous models of salmon survival, we applied a random effects model to a rich dataset of over 285,000 individually-tagged Snake River spring/summer Chinook salmon between 2000 and 2015.

Reviewer #2:

 The term multivariate seems to be used in the text to refer to a model with multiple predictor variables. Strictly speaking, multivariate refers to a model with more than one response variable. However, the authors only have one response variable in their model (SAR), so the term multivariate should not be used.

We have removed multivariate from the paper.

 Page 9, equation 2 and associated text. If beta is a vector, please present in in bold. The term x_t should be matrix (where rows are measurements and columns are the different predictors), not a vector as described in the text. The text should be corrected and the term should be represented as a bold capital X. Correction should be applied to Table 3 as well.

We have corrected the equations throughout as per the suggestions of the reviewer and the editor

 Page 11, first paragraph. The text implies that SAR (s_rjt) is binomially distributed. Actually, s_rjt is a parameter of the binomial distribution. What is binomially distributed is the number of survivors (k_rjt). Please revise.

The sentence now reads,

Given the total number of juveniles that migrated downstream n_jt and the predicted SAR s_jt, the number of juvenile fish k_jt that survived to adulthood and were detected at one of the eight main-stem on the Columbia River and Snake River is binomially distributed

 The authors frequently refer to the top model(s) as the models that had best-fit. AIC is not simply a measure of fit but rather it is a statistics that represents a balance between model fit and complexity. Therefore, it's not appropriate to refer to the selected model(s) as the model(s) of best-fit (a model with best fit may not even be included among the selected models). I suggest referring to selected models as more parsimonious or optimal model(s).

We have removed best-fit throughout and changed the text to use the term parsimonious.

 Page 16. The authors refer to the intervals as credible intervals. The term credible is used for interval determined using Bayesian inference, but it seems the authors are using a frequentist inference framework. So, to avoid confusion I suggest they refer to the interval as confidence interval throughout the paper.

We have changed the text to confidence interval throughout.

 To my knowledge, marginal AIC should not be used to conduct model selection in linear mixed models in the way it was conducted by the authors (i.e. comparing models with different fixed and random effects structures). I suggest the authors check out the paper by Vaida & Blanchard (2005) and the book by Zuur et al. (2009). In the latter, the authors suggest to first select the optimal random effects structure by applying AIC to models with all fixed effects but different random effects. Once the optimal random structure is selected, AIC can then be used to compare models that have different fixed effects but the same (previously selected) optimal random effects structure.

We have considered reviewer 2’s #6 comments carefully. We see their argument as containing to two parts. 

 Vaida is one of many papers about conditional AIC, which should minimize predictive variance for data given predicted random effects. The marginal AIC that we use minimizes predictive variance for new random effects. 

 Zuur suggests selecting random effects given specified fixed-effect structure when using REML. We are not using a restricted maximum likelihood, so again we feel that mAIC is appropriate.

Additionally, we recognize that this was an important consideration for the editor and the reviewer, so we included the following text in the discussion where we addressed model fit. We are happy to include this explanation in the paper:

We recognize that there are multiple ways to evaluate model fit and specification (i.e., fixed and random effects structures) for mixed effects models. For instance, Vaida and Blanchard [46] propose using conditional AIC, because marginal AIC tends to favor smaller models with fewer random effects [47]; however, conditional AIC is not computationally reasonable for large datasets such as ours [47]. Zuur [48] proposed, in the case of REML models (restricted maximum likelihood), selecting the number random effects using marginal AIC conditioned on all of the fixed effects, and then choosing the fixed effects conditional on the structure of the optimal random effects. Because we have not implemented REML in our joint likelihood, and given that we have greater than 1600 observations [47], we have chosen to compare models using the marginal AIC and recognize that we may be underestimating the optimal number of random effects. Future analyses may also consider using conditional AIC approaches (e.g., the DHARMa package in R [49]) to evaluate model misspecification for a larger suite of random effects models; however, in this paper we have specifically focused on evaluating the effects of the unknown processes for day and year.

 It seems Figs. 4 and 5 were uploaded in the wrong order. Please check.

We checked and they were uploaded in the correct order.

Brandon Chasco and others

---

## [Editor Report · Decision Letter 2]

4 Jan 2021

PONE-D-20-13616R2

Differential impacts of freshwater and marine covariates on wild and hatchery Chinook salmon marine survival

PLOS ONE

Dear Dr. Chasco,

Thank you for submitting your manuscript to PLOS ONE. After careful consideration, we feel that it has merit but does not fully meet PLOS ONE’s publication criteria as it currently stands. Therefore, we invite you to submit a revised version of the manuscript that addresses the points raised during the review process.

Below are a few minor revisions that I invite you to consider:

Line 91. I suggest adding the words  “structure in “ after inherent. By including autocorrelation and random effects, the model is providing some structure to the variability, e.g.,  deviations this year is related to deviations last year, and year and day effects on SAR are drawn from common random distributions (Eqns. 4 and 5, respectively).

Line 194, Suggest revising to “Where the elements of matrix **Σ**, ...” for clarity.

Eqn. 7. This equation makes more sense to me in the original form: sig^2/(1-rho^2) x rho^delta, which is more consistent with equations 4 and 5. Although they are equivalent, the original form might be more intuitive to readers. (Same for the equation on line 195).

Table 5, Suggest adding subscripts 1 and 2 (or PDO.sum and CUI.spr as in Fig. 6) to the beta terms (the effects of first and second marine covariates, respectively). I assume these are elements of the vector **β**. Also, in this table, I suggest aligning the Parameter description with the Symbol. It looks like they are currently offset by 1 line.

Yes, I recommend including the suggested paragraph on conditional vs marginal AIC.

We look forward to receiving your revised manuscript. Wishing you the best for the New Year.

Kind regards,

Carrie A. Holt

Academic Editor

PLOS ONE

---

## [Author Response · Author response to Decision Letter 2]

19 Jan 2021

PONE-D-20-13616R1

Differential impacts of freshwater and marine covariates on wild and hatchery Chinook salmon marine survival

PLOS ONE

Below are a few minor revisions that I invite you to consider:

Line 91. I suggest adding the words “structure in “ after inherent. By including autocorrelation and random effects, the model is providing some structure to the variability, e.g., deviations this year is related to deviations last year, and year and day effects on SAR are drawn from common random distributions (Eqns. 4 and 5, respectively).

We have added the suggested words to the text.

Line 194, Suggest revising to “Where the elements of matrix Σ, ...” for clarity.

Eqn. 7. This equation makes more sense to me in the original form: sig^2/(1-rho^2) x rho^delta, which is more consistent with equations 4 and 5. Although they are equivalent, the original form might be more intuitive to readers. (Same for the equation on line 195).

We have added the following text that combines the comments related to the text and equation 7,

Σ_(j,j+δ)=σ^2 〖ρ 〗^δ/(1-〖ρ 〗^2 ) (7)

Where the elements of the covariance matrix (Σ) are a function of the variance parameter σ^2 which is rescaled based on the correlation ρ between days and the δ number of days between observations.

Table 5, Suggest adding subscripts 1 and 2 (or PDO.sum and CUI.spr as in Fig. 6) to the beta terms (the effects of first and second marine covariates, respectively). I assume these are elements of the vector β. Also, in this table, I suggest aligning the Parameter description with the Symbol. It looks like they are currently offset by 1 line.

Effect of first marine covariate1 β_1 0.488 ( 0.228, 0.747 )2 0.478 ( 0.218, 0.737 )2 0.458 ( 0.22, 0.695 )3

Effect of second marine covariate1 β_2 0.547 ( 0.283, 0.812 )2 0.56 ( 0.295, 0.825 )2 -0.608 ( -0.82, -0.396 )3

1The first and second marine covariates are elements of the covariate vector β.

2For hatchery spring/summer Chinook salmon, the first and second marine covariates in the model with the lowest AIC were transport.sum and npgo.sum, respectively.

3For wild spring/summer Chinook salmon, the first and second marine covariates in the model with the lowest AIC were cui.spr and pdo.sum, respectively.

Yes, I recommend including the suggested paragraph on conditional vs marginal AIC.

---

## [Editor Report · Decision Letter 3]

25 Jan 2021

Differential impacts of freshwater and marine covariates on wild and hatchery Chinook salmon marine survival

PONE-D-20-13616R3

Dear Dr. Chasco,

We’re pleased to inform you that your manuscript has been judged scientifically suitable for publication and will be formally accepted for publication once it meets all outstanding technical requirements.

Kind regards,

Carrie A. Holt

Academic Editor

PLOS ONE
---

## [Editor Report · Acceptance letter]

29 Jan 2021

PONE-D-20-13616R3 

Differential impacts of freshwater and marine covariates on wild and hatchery Chinook salmon marine survival 

Dear Dr. Chasco:

I'm pleased to inform you that your manuscript has been deemed suitable for publication in PLOS ONE. Congratulations! Your manuscript is now with our production department. 

Kind regards, 

on behalf of

Dr. Carrie A. Holt 

Academic Editor

PLOS ONE